# Risk Factors Attributable to Hypertension among HIV-Infected Patients on Antiretroviral Therapy in Selected Rural Districts of the Eastern Cape Province, South Africa

**DOI:** 10.3390/ijerph191811196

**Published:** 2022-09-06

**Authors:** Urgent Tsuro, Kelechi E. Oladimeji, Guillermo-Alfredo Pulido-Estrada, Teke R. Apalata

**Affiliations:** 1Department of Public Health, Faculty of Health Sciences, Walter Sisulu University, Mthatha 5100, South Africa; 2Department of Laboratory Medicine and Pathology, Faculty of Health Sciences, Walter Sisulu University, Mthatha 5100, South Africa; 3College of Graduate Studies, University of South Africa, Pretoria 0001, South Africa

**Keywords:** anti-retroviral therapy, HIV/AIDS, hypertension prevalence, hypertension risk factors, hypertension treatment

## Abstract

Background: Antiretroviral therapy has improved HIV patients’ quality of life and life expectancy. However, complications have emerged in the form of hypertension. In the rural Eastern Cape, there is minimal information about HIV-infected people. The current study intended to evaluate the factors associated with hypertension in HIV-infected individuals receiving antiretroviral therapy in rural areas of South Africa’s Eastern Cape. Methods: For this cohort study, HIV-positive people taking antiretroviral therapy aged 15 and up were recruited at random from several rural locations in the Eastern Cape. Using Cox univariate and multivariate analyses, the key predictors of hypertension were found. Results: Of the total participants (n = 361), 53% of individuals had hypertension. In the Cox multivariate model, patients that had hypertension heredity, BMI ≥ 25 kg/m^2^, eGFR < 60 mL/min/1.73 m^2^, advanced and severe CD4 counts, 1TFE and 1T3E regimens, and the male gender were found to be at greater risk of hypertension. Conclusions: The findings of this study indicate that hypertension is a prevalent concern among HIV patients receiving antiretroviral therapy. HIV patients should have their blood pressure checked regularly, and they should be screened for high blood pressure and given treatment for it.

## 1. Introduction

As antiretroviral therapy (ART) was introduced, the life expectancy of people living with HIV (PLHIV) increased [1]. PLHIV on ART now have reduced mortality rates due to improved ART, as it suppresses HIV replication [2]. Therefore, PLHIV are progressively ageing and now facing new problems in the form of non-communicable diseases (NCDs), including hypertension [3]. One billion individuals globally are affected by hypertension, which qualifies it to be a global public health concern [4]. According to research, raised blood pressure claims approximately 15 million lives annually [5]. Reports suggest that the world’s highest age-adjusted hypertension prevalence is found in sub-Saharan Africa (SSA) [6]. The incidence of hypertension in SSA is 1.5 times that of Europe, the United States, and Australia [7]. In 2017, it was projected that 80% of the global deaths attributable to hypertension would occur in low- and middle-income countries (LMICs) [5]. As such, the incidence and prevalence of high blood pressure is escalating in LMICs, with the largest increase in SSA [5], and South Africa having the worst burden [8].

South Africa, a middle-income country struggling with the worst HIV epidemic globally [9], now encounters a superfluous challenge of increasing NCDs, mostly hypertension. There is a dearth of incidence estimates for hypertension, but research suggests thatthe prevalence of hypertension is on an increase in South Africa [10]. The prevalence of hypertension in the general population was about 15% in the late 1990s [11,12,13]. In 2008, Bärnighausen et al. reported a prevalence estimate of 33% among South African adults 15–50 years of age [14], while Berry et al. in 2017 estimated an adult hypertension prevalence of 35% for the same country [15]. As reported by the 2016 Demographic Health Survey, hypertension prevalence among South African females and males was 46.0% and 44.0%, respectively [16].

Treatment non-compliance, urbanisation, and behavioural risk factors such as poor nutrition, physical inactivity, and alcohol and cigarette use have all been linked to higher than world average prevalence in numerous LMICs [17]. PLHIV on ART are at a bigger risk of hypertension than the HIV uninfected [18]. Traditional hypertension risk factors explain less of the worsened risk among PLHIV [19,20,21]. Elevated hypertension prevalence in PLHIV is possibly caused by chronic inflammation, renal disease, ART exposure, and higher levels of behavioural risk factors [18]. Several risk variables, including advanced age, a family history of hypertension, male gender, a high viral load, a longer duration of HIV infection, and a high body mass index (BMI), may contribute to the greater incidence of hypertension in PLHIV [19,21]. More than 50% of ART-experienced individuals that are above 50 years old have hypertension [22].

South Africa’s ART programme has expanded to be the largest globally, with an estimated 4.8 million adults on ART in 2018 [23]. While South Africa continues to be one of the furthermost unequal societies [24], the country has experienced swift urbanization and income growth leading to lifestyle, stress and dietary changes among its citizens [25]. As sedentary lifestyles and obesity increase, the prevalence of hypertension is likely to rise even quicker in the coming years [25]. Even though certain behavioural risk factors may contribute to a higher hypertension prevalence among PLHIV, those receiving ART also experience immune activation, which creates a higher risk for hypertension [26,27]. As more PLHIV are on ART and live normal life spans, hypertension is likely to cause proportionally greater morbidity and mortality [28,29]. Management of PLHIV needs multidisciplinary team work, taking into account the start of NCDs, premature ageing, and an increased risk of drug-to-drug interactions and drug toxicities due to polypharmacy [30]. In addition, for HIV-infected individuals using antiretrovirals (ARVs) such as abacavir (ABC), Lamivudine (3TC) and Zidovudine (AZT) [31], the risk of developing hypertension keeps increasing [32]. Furthermore, TDF exposure has been linked to a faster decline in eGFR in the first 6–12 months after starting ART in African HIV patients with minimal or moderate renal dysfunction at baseline [33].

Although few studies have investigated hypertension and its associated risk factors in some rural areas in South Africa, we are not aware of studies that have sought to determine the factors associated with hypertension among PLHIV in rural Eastern Cape, South African adults. Therefore, the purpose of this study was to investigate the factors associated with hypertension among PLHIV on ART in selected rural districts of the Eastern Cape.

## 2. Materials and Methods

### 2.1. Study Design and Setting

The present cohort study included the black ethnicity group, serviced by health facilities in selected districts of the Eastern Cape Province, South Africa. The Eastern Cape Province has a total population of approximately 7 million people and is the third most populous province in South Africa [34].

### 2.2. Eligibility Criteria

PLHIV on ART and at least 15 years of age were allowed to take part in this study. Pregnant individuals, patients with a history of developing hypertension, and those reluctant to take part were excluded from the study.

### 2.3. Ethical Approval

The study was in line with the ethical guidelines as declared by Helsinki [35], and approval was obtained from Walter Sisulu University ethics committee, protocol number (048/2019) and the Eastern Cape department of Health (EC_201907_020). Before completing the written informed consent, potential participants were issued with an information sheet in English and their vernacular. The information sheet contained the process of research, the rights of the participants, as well as the contact person’s information.

### 2.4. Sample Size and Sampling

The suitable sample size was calculated at a 95% confidence level using the formula stated below: $$n=\frac{p\left(1-p\right)z^{2}}{d^{2}}=\frac{0.31\left(1-0.31\right)1.96^{2}}{0.05^{2}}=329$$
where *n* is the number of participants, *z* is the score from the normal distribution table using 95% confidence level, *p* is the expected proportion of patients with hypertension, *p* was 0.31 [36], d is the margin of error, which is 5%. 

After adding 20%, 390 participants were included in the study.

### 2.5. Sampling Procedure

Three hundred and ninety HIV-positive individuals were enrolled randomly from selected districts of the Eastern Cape.

### 2.6. Data Collection

The World Health Organization (WHO) Stepwise questionnaire and consent forms were digitalised into Research Electronic Data Capture (RedCap), a web-based online survey tool [37]. The digitalised tool was then utilised for the face-to-face interviews conducted by trained research staff to obtain informed consent and data. A registered nurse carried out all invasive procedures.

### 2.7. Definitions

#### 2.7.1. Hypertension

Trained healthcare professionals performed standardized blood pressure measurements using electronic monitors. Before measuring their blood pressure, participants were instructed to rest for at least five minutes. Three measurements were obtained at 5 min intervals, and their average was determined thereafter. The Eighth Report of the Joint National Committee on the Prevention, Detection, Evaluation, and Treatment of High Blood Pressure (JNC-8) was utilized to identify individuals with high blood pressure [38]. Incident hypertension was defined as average systolic blood pressure (SBP) ≥ 140 mmHg and/or average diastolic blood pressure (DBP) ≥ 90 mmHg.

#### 2.7.2. Assessment of Overweight/Obesity

On a typical beam balance, participants’ weight was measured to the nearest 0.1 kg, and their height was measured to the nearest 0.1 cm on a mounted Stadiometer. BMI was determined, and subjects were then classified according to WHO guidelines. [39]. BMI was categorized into two groups: <25 and ≥25 kg/m^2^.

#### 2.7.3. Immunological Status

The patient’s blood was collected by a registered nurse and taken to the laboratory for HIV-related tests such as status and viral load. HIV status was categorised as positive or negative. Viral load was categorised as <50 copies/mL, 50–1000 copies/mL, >1000 copies/mL.

#### 2.7.4. Renal Function

Estimated glomerular filtration rate (eGFR) was used to assess renal function. Following the chronic kidney disease criteria (eGFR < 60 mL/min/1.73 m^2^) from the KDIGO 2012 Clinical Practice Guideline for the Evaluation and Management of Chronic Kidney Disease [40], the resulting data were categorized. 

#### 2.7.5. Socioeconomic and Environmental Variables

Information about sociodemographic and environmental variables was collected during personal face-to-face interviews using a WHO stepwise questionnaire uploaded on RedCap. Age (years) was considered as a continuous variable. All participants were black; gender, smoking status, alcohol consumption, and hypertension heredity were recorded. Hypertension heredity considered a brother, sister, father or mother of the participant. Self-reported levels of physical activity were measured and categorized based on the WHO recommendations [41].

### 2.8. Data Analysis

R studio version 4.2.0 was used for data analysis. A Cox proportional hazard model was used for univariate analysis, and all variables with a *p* value < 0.2 were identified and included in the multivariate model. All the identified variables qualified to be included in the multivariate Cox model, and all variables with a *p* value < 0.05 were identified as statistically significant at 95%. The final model was then tested using the likelihood ratio test to assess the contribution of the variables, and concordance was considered for the model’s predictive power where the *p* value was found to be >0.05. The Schoenfeld residual global test was utilised to test the proportionality assumption, and the global *p* value was found to be >0.05. Kaplan–Meier curves were constructed for the significant variables.

### 2.9. Validity and Reliability

To ascertain validity and reliability, a pilot study was conducted on 180 participants, and the data were excluded from this study’s analysis.

## 3. Results

### 3.1. Characteristics of the Study Population

A total of 390 HIV positive individuals were potentially eligible to take part in the study. Of those, 29 participants were excluded from the analysis, as they were not on ART. Thus, 361 individuals qualified to be in the study according to the pre-defined inclusion criteria, and of the 361 study participants, 191 (53%) were hypertensive; Figure 1.

### 3.2. Characteristics of the Study Population

Table 1 shows the socio-demographic and clinical characteristics of the study participants. The median age of the study participants was 51 years (IQR 39.0–59.0), and 88.9% of the participants were of the female gender. Participants that smoked tobacco were 8.3%, and those that consumed alcohol were 11.9%. A greater proportion of the participants (58.2%) were taking part in at least one physical activity, and 63.7% had a BMI greater than or equal to 25 kg/m^2^. Furthermore, 40.2% of the participants had a close family member who had a history of developing hypertension. In terms of HIV-related characteristics, the bulk of the participants, 74.8%, had viral loads of less than 50 copies/mL, and viral loads greater than 1000 copies/mL were the least populated group at 6.9%. A greater proportion of the participants (59.3) had mild CD4, with most of the participants 73.4% were on 1TFE. The majority of the participants, 58.7%, had less than 5 years on ART Table 1.

### 3.3. Risk Factors Associated with the Development of Hypertension

All variables that had a *p* value less than 0.20 in the univariate analysis qualified to be in the multivariate model; Table 2. Patients who had hypertension heredity (adjusted hazard ratio (aHR): 1.66, 95% CI 1.21–2.27, *p* = 0.0015), BMI ≥ 25 (kg/m^2^) (aHR: 1.97, 95% CI 1.35–2.87, *p* = 0.0005), eGFR ≥ 60 (aHR: 0.54, 95% CI 0.36–0.71, *p* = 0.0000) and male gender (aHR: 2.83, 95% CI 1.76–4.56, *p* = 0.0000) were identified as independent risk factors of hypertension in our adjusted Cox regression model; Table 2. HIV-related variables such as advanced CD4 count (aHR: 2.28, 95% CI 1.50–3.45, *p* = 0.0001), severe CD4 count (aHR: 2.48, 95% CI 1.71–3.61, *p* = 0.0000), ART regimens 1T3E (aHR: 3.39, 95% CI 1.62–7.09, *p* = 0.0012) and 1TFE (aHR: 5.99, 95% CI 3.19–11.27, *p* = 0.0000) were also identified as risk factors associated with the development of hypertension; Table 2.

### 3.4. Impact of the Identified Risk Factors on the Incidence of Hypertension

Participants that had a stronger renal function were identified as having a lesser risk of hypertension than those with a weaker renal function. The variables that had an association with hypertension incidence were participants that had hypertension heredity, BMI ≥ 25 kg/m^2^, ART regimens 1TFE and 1T3E, advanced and severe CD4 counts, and those of the male gender were at a higher risk of developing hypertension, as they lie on the right side of the reference line. The model had a greater predictive power of 77%; Figure 2.

### 3.5. Kaplan–Meier Curves

The Kaplan–Meier estimates for the survival probability of remaining hypertension free due to the identified six significant risk factors are shown in Figure 3. With a lower median time and survival, probability in (c), BMI ≥ 25 kg/m^2^ was implicated in causing hypertension. Patients with renal function < 60 had a greater likelihood of being hypertensive according to (d). Individuals that had a close family member with history of developing hypertension had a high chance of being hypertensive with a lower median time as depicted in (b). According to (a), the male gender had a greater likelihood of developing hypertension with a median time less than that of women; Figure 3. ART regimen 1TFE had a lower mean time median time compared to the other drugs, while 1S3E had a greater survival probability in (e). Finally, (f) showed that severe CD4 counts had a shorter mean time to develop hypertension compared to the other categories.

## 4. Discussion

In South Africa, hypertension is a recognized health risk factor that contributes to the majority of vascular illnesses [42,43]. According to some research, the prevalence of hypertension is rising more rapidly in emerging countries than it is in industrialized ones [44,45]. In addition, evidence shows that PLHIV on ART have significantly more NCD risk factors than individuals who are not on ART [46]. The immunological activation and inflammation caused by HIV may enhance the CVD risk for PLHIV, as well as dyslipidaemia, insulin resistance, and glucose intolerance brought on by ART therapy [47,48]. As South Africa’s ART program is increasing, there is an epidemiological alteration from HIV related to NCDs such as hypertension [23]. Therefore, this study sought to determine the risk factors associated with the incidence of hypertension among PLHIV on ART in the rural Eastern Cape, South Africa.

In the present study, we established a hypertension prevalence of 53% among PLHIV on ART in the rural Eastern Cape, South Africa, which was consistent with findings from a study in the Eastern Cape [42]. Remarkably, this prevalence is approximately 4.6-fold that reported in a related study conducted in rural Tanzania [49]. This prevalence is higher than that of South Africa, which is between 5% and 52% [50]. Nevertheless, the prevalence in the current study confirms that there is a hypertension burden in this rural population of South Africa [51,52,53].

In the current study, we established that patients with close family members that had a history of developing hypertension, BMI ≥ 25 kg/m^2^, eGFR < 60 mL/min/1.73 m², and belonging to the male gender had a greater risk of hypertension. This was consistent with the findings of a study conducted in Tanzania where BMI ≥ 25 kg/m^2^ and eGFR less than 60 were found to be stronger risk factors of hypertension among HIV-infected individuals on ART [49]. Advanced and severe CD4 counts as well as ART regimens 1T3E and 1TFE were also implicated in causing hypertension in this population. Our findings were similar to those found in some studies conducted in Africa [20,54,55].

With an AHR of 2.83, the male gender was more than twice as likely to develop hypertension than females. This variance in gender could be clarified using hormonal dissimilarities that shield women from hypertension. In addition, men are known to live a more sedentary lifestyle than women [56]. Moreover, the effect of gender on the risk of being hypertensive is not well reported, as there are contradictory results being stated regarding the association of gender and the prevalence of hypertension. As such, Berhane et al. [57], Alberts et al. [58] and Mkhonto et al. [59] reported a high hypertension prevalence among HIV positive women. However, the current study is contrary to these findings, as it reports a greater prevalence of hypertension among males than females. Several studies conducted in different districts of South Africa reported a lower hypertension prevalence among females [60,61,62,63]. 

The BMI ≥ 25 kg/m^2^ group was 97% more likely to develop hypertension compared to the BMI < 25 kg/m^2^. The relationship between hypertension and increasing BMI has previously been reported [53,64,65]. The threat of hypertension was reported to have increased by 49% for every five-unit increase in BMI [66]. BMI is an essential modifiable risk factor, yet improving healthy lifestyle to reduce BMI has not been given priority in lessening of hypertension in developing countries [67]. Weight reduction, exercise, and eating healthy are required to reduce overweight and obesity among PLHIV [68].

Participants that had a family member with a history of hypertension were 66% more vulnerable to developing hypertension. This was consistent with other study findings [69]. Similarly, another study reported that HIV-infected patients with a family history of hypertension were more likely to develop hypertension when on ART [70]. 

This study identified ART regimens 1T3E and 1TFE as significant risk factors of hypertension. This was consistent with findings from some studies carried out among PLHIV [71,72]. Advanced and severe CD4 counts were statistically significant with hypertension. This was in agreement with findings from other studies that identified low CD4 count as a significant factor [54,73]. However, these findings were different from those reported by Njelekela et al. [74]. In communities that are HIV positive, hypertension is quite frequent and may even be more prevalent than in groups that are HIV negative [21].

In this study, renal insufficiency was also significantly associated with an increased risk of hypertension. Participants with strong renal function were 46% less vulnerable to developing hypertension. An independent association between renal insufficiency and hypertension in PLHIV on ART was reported in other related studies [75,76]. 

In order to prevent organs from being damaged, it is critical to control hypertension among PLHIV, thus reducing mortality [77,78]. The findings of this study have implications for clinical practice. Identifying the risk factors associated with hypertension among PLHIV provides up-to-date evidence to understand the prognosis of hypertension in this population. It is also critical to develop guidelines for regular blood pressure monitoring for PLHIV receiving ART to reduce further complications and mortality. Moreover, identifying the predictors could help clinicians prioritize and consider their routine clinical practice for PLHIV. Early identification followed by adequate therapeutic care could help reduce the morbidity associated with this condition. The clinical effects of hypertension can be severe and are related to target organ damage [79].

## 5. Conclusions

The significant need for integrating hypertension screening and care in rural Eastern Cape communities is highlighted by the high incidence of hypertension among PLHIV on ART in this study. A public health approach would be significantly more successful because primary health care is frequently inaccessible to those in need in many sections of the nation, especially rural areas. Identifying and managing modifiable risk factors, screening for common NCDs, and diagnosing, treating, monitoring, and, when necessary, referring patients with common NCDs such as hypertension are highly recommended [80]. When HIV and NCDs prevention and care are integrated, the missed opportunities to practice hypertension prevention are reduced. 

This research had certain limitations, such as the fact that males were under-represented, which prevented us from gaining a thorough picture of the gender-based contribution to hypertension prevention, as well as the inability to identify their special requirements. Furthermore, since this was a cohort study, it was not able to identify causal correlations. Consequently, additional studies are required to evaluate the progression of hypertension and its care in the same environment. Furthermore, future studies should consider certain restrictions. Given that the number of PLHIV on ART with high blood pressure is on the rise, an appropriate and innovative intervention is required.

## Figures and Tables

**Figure 1 ijerph-19-11196-f001:**
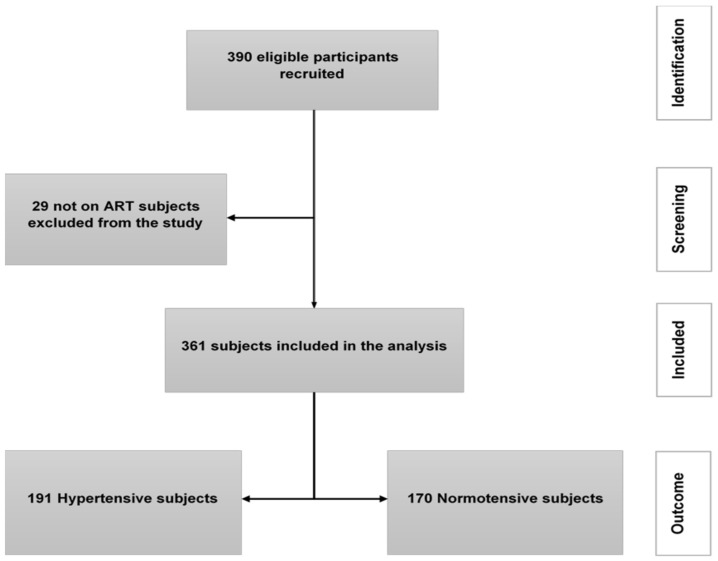
Patient flow chart of the study profile.

**Figure 2 ijerph-19-11196-f002:**
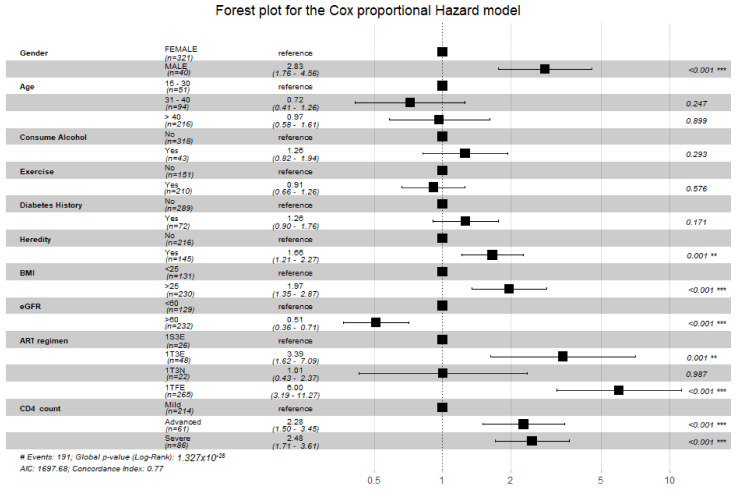
Forest plot for the Cox proportional hazard model. BMI: body mass index; eGFR: estimated glomerular filtration rate; ART: antiretroviral therapy; HR: hazard ratio; CI: confidence interval; WHO: World Health Organization; **: <0.001; ***: <0.0001.

**Figure 3 ijerph-19-11196-f003:**
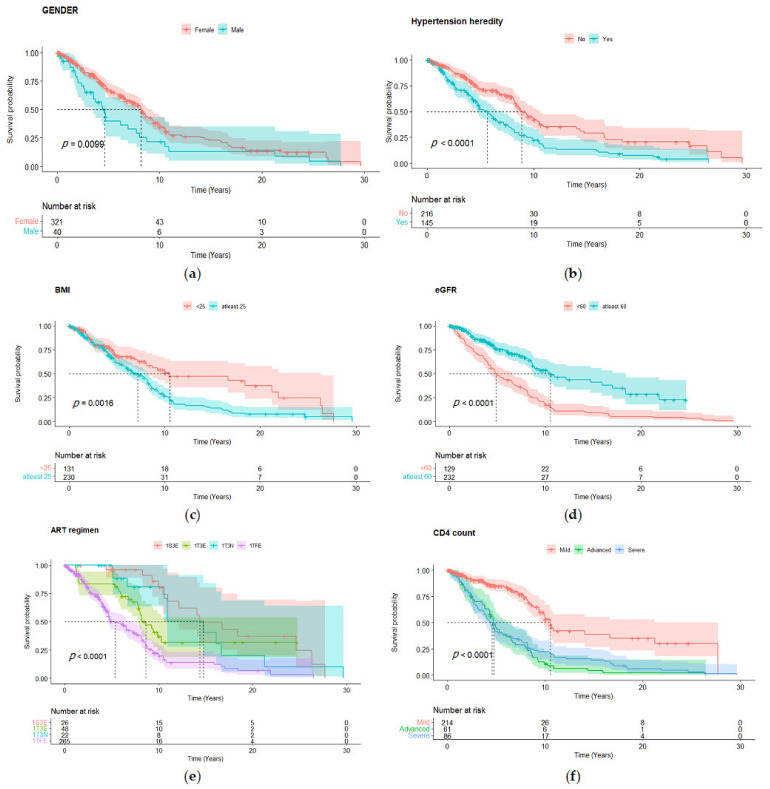
Kaplan–Meier curve and risk tables of hypertension development based on: (**a**) gender, (**b**) hypertension heredity, (**c**) BMI, (**d**) eGFR, (**e**) ART regimen, (**f**) CD4 count. BMI: body mass index; eGFR: estimated glomerular filtration rate; ART: antiretroviral therapy; HR: hazard ratio; CI: confidence interval; WHO: World Health Organisation.

**Table 1 ijerph-19-11196-t001:** Summary of patient characteristics.

	Hypertensive	Normotensive	Total
**Age, Median (IQR)**	51 (39.0–59.0)	43 (33.0–54.5)	45 (35–56)
**15–30 years, n (%)**	+20 (10.5)	31 (18.2)	51 (14.1)
**31–40 years, n (%)**	42 (22.0)	52 (30.6)	94 (26.0)
**35–45 years, n (%)**	129 (67.5)	87 (30.6)	216 (59.8)
**Gender, n (%)**			
**Female**	160 (83.8)	161 (94.7)	321 (88.9)
**Male**	31 (16.2)	9 (5.3)	40 (11.1)
**Current smokers, n (%)**			
**Non-Smoker**	166 (86.9)	165 (97.1)	331 (91.7)
**Smoker**	25 (13.1)	5 (2.9)	30 (8.3)
**Consume alcohol, n (%)**			
**No**	161 (84.3)	157 (92.4)	318 (88.1)
**Yes**	30 (15.7)	13 (7.2)	43 (11.9)
**Exercise, n (%)**			
**No**	110 (57.6)	41 (24.1)	151 (41.8)
**Yes**	81 (42.4)	129 (75.9)	210 (58.2)
**Hypertension heredity, n (%)**			
**No**	90 (47.1)	126 (74.1)	216 (59.8)
**Yes**	101 (52.9)	44 (25.9)	145 (40.2)
**Diabetes history, n (%)**			
**No**	137 (71.7)	152 (89.4)	289 (80.1)
**Yes**	54 (28.3)	18 (10.6)	72 (19.9)
**BMI categories, n (%)**			
**<25 (kg/m^2^)**	48 (25.1)	87 (48.8)	131 (36.3)
**≥25 (kg/m^2^)**	143 (74.9)	83 (51.2)	230 (63.7)
**eGFR, n (%)**			
**<60 mL/min/1.73 m²**	123 (64.4)	6 (3.5)	129 (35.7)
**≥60 mL/min/1.73 m²**	68 (35.6)	164 (96.5)	232 (64.3)
**VL category, n (%)**			
**<50 (copies/mL)**	139 (72.8)	131 (77.1)	270 (74.8)
**50–1000 (copies/mL)**	28 (14.7)	38 (22.4)	66 (18.3)
**>1000 (copies/mL)**	24 (12.6)	1 (0.6)	25 (6.9)
**CD4 count, n (%)**			
**Mild**	53 (27.7)	161 (94.7)	214 (59.3)
**Advanced**	56 (29.3)	5 (2.9)	61 (16.9)
**Severe**	82 (42.9)	4 (2.4)	86 (23.8)
**ART regimen, n (%)**			
**1S3E**	14 (7.3)	12 (7.1)	26 (7.2)
**1T3E**	26 (13.6)	22 (12.9)	48 (13.3)
**1T3N**	11 (5.8)	11 (6.5)	22 (6.1)
**1TFE**	140 (73.3)	125 (73.3)	265 (73.4)
**ART duration, n (%)**			
**<5 years**	106 (55.5)	106 (62.4)	212 (58.7)
**5–10 years**	66 (34.6)	53 (31.2)	119 (33.0)
**>10 years**	19 (9.9)	11 (6.2)	30 (8.3)

BMI: body mass index; eGFR: estimated glomerular filtration rate; ART: antiretroviral therapy; n: number of patients; VL: viral load; WHO: World Health Organisation; IQR: interquartile range. 1S3E: Stavudine (D4T) + Lamivudine (3TC) + Efavirenz (EFV), 1T3E: Tenofovir (TDF) + Lamivudine (3TC) + Efavirenz (EFV), 1T3N: enofovir (TDF) + Lamivudine (3TC) + Nevirapine (NVP), 1TFE: Tenofovir (TDF) + Emtricitabine (FTC) + Efavirenz (EFV) [31].

**Table 2 ijerph-19-11196-t002:** Cox regression analysis of risk factors of hypertension.

	Univariate Model		Multivariate Model	
	Unadjusted HR (95% CI)	*p* Value	Adjusted HR (95% CI)	*p* Value
**Gender MALE**	1.66 (1.13–2.45)	0.0106	2.83 (1.76–4.56)	0.0000
**Age 31–40 (years)**	0.96 (0.56–1.64)	0.8790	0.72 (0.41–1.26)	0.2467
**Age > 40 years**	1.26 (0.79–2.03)	0.3330	0.97 (0.58–1.61)	0.8991
**Consume alcohol Yes**	1.51 (1.02–2.23)	0.0414	1.26 (0.82–1.94)	0.2932
**Exercise Yes**	0.62 (0.47–0.83)	0.0015	0.91 (0.66–1.26)	0.5756
**Diabetes history Yes**	1.39 (1.01–1.91)	0.0422	1.26 (0.90–1.76)	0.1706
**Hypertension heredity Yes**	1.93 (1.45–2.57)	0.0000	1.66 (1.21–2.27)	0.0015
**BMI ≥ 25 kg/m^2^**	1.69 (1.21–2.34)	0.0019	1.97 (1.35–2.87)	0.0005
**eGFR ≥ 60 mL/min/1.73 m²**	0.38 (0.29–0.52)	0.0000	0.54 (0.36–0.71)	0.0000
**ART regimen 1T3E**	1.93 (0.99–3.73)	0.0515	3.39 (1.62–7.09)	0.0012
**ART regimen 1T3N**	1.20 (0.53–2.71)	0.6633	1.01 (0.43–2.37)	0.9868
**ART regimen 1TFE**	3.92 (2.22–6.93)	0.0000	5.99 (3.19–11.27)	0.0000
**CD4 count Advanced**	3.74 (2.57–5.45)	0.0000	2.28 (1.50–3.45)	0.0001
**CD4 count Severe**	3.10 (2.19–4.39)	0.0000	2.48 (1.71–3.61)	0.0000

BMI: body mass index; eGFR: estimated glomerular filtration rate; ART: antiretroviral therapy; HR: hazard ratio; 1S3E: Stavudine (D4T) + Lamivudine (3TC) + Efavirenz (EFV), 1T3E: Tenofovir (TDF) + Lamivudine (3TC) + Efavirenz (EFV), 1T3N: enofovir (TDF) + Lamivudine (3TC) + Nevirapine (NVP), 1TFE: Tenofovir (TDF) + Emtricitabine (FTC) + Efavirenz (EFV) [31]; CI: confidence interval; WHO: World Health Organization.

## Data Availability

Data will be made available upon request from the corresponding author.

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
