# Peer review of "Risk Factors Attributable to Hypertension among HIV-Infected Patients on Antiretroviral Therapy in Selected Rural Districts of the Eastern Cape Province, South Africa"

_ijerph, 2022, doi:10.3390/ijerph191811196_

Round 1

Reviewer 1 Report

Overall good manuscript with evidence of increase in hypertension among PLHIV. Comments are below:

1. With hypertension being a common disease among the general population and not specific to PLHIV, are the health risks related to hypertension greater for PLHIV? Also, many of the risk factors that were found to be significant are already known risk factors for hypertension. However, the ART effect is interesting and is specific to PLHIV. It would be nice for the authors to elaborate more on this. Also, would the ART duration be found significant if it was separated into ART regimens? 

2. The authors suggest integrating hypertension screening. Is this not part of standard routine care for individuals in Eastern Cape? If it is, how would care differ? 

3. A BMI > 25 kg/m2 was found to be a risk factor for the development of hypertension and the authors further suggest weight reduction, exercise, and diet as part of obesity reduction; however, exercise was not found to be significant factor. What are the authors thoughts on this? 

4. With the males being underrepresented, could there be sampling error associated with the gender analysis? 

5. Could kidney disease result from hypertension rather than kidney disease being a risk factor for hypertension?

6. There are many reference errors throughout the manuscript that is listed as 'Error! Reference source not found'. Please ensure that all references are accurately noted. 

Author Response

1. With hypertension being a common disease among the general population and not specific to PLHIV, are the health risks related to hypertension greater for PLHIV? Also, many of the risk factors that were found to be significant are already known risk factors for hypertension. However, the ART effect is interesting and is specific to PLHIV. It would be nice for the authors to elaborate more on this. Also, would the ART duration be found significant if it was separated into ART regimens? 

Author's response: We thank you for the comments. We made revision to highlight evidence which reports the increased burden of CVDs like hypertension particularly among those with HIV. Please lines 243-257 in pages 9 & 10. We applied duration of ART in the cox regression model which shows the differing levels of risk factors for hypertension according to the ART regimen. Please see Table 2.

2. The authors suggest integrating hypertension screening. Is this not part of standard routine care for individuals in Eastern Cape? If it is, how would care differ? 

Author's response: We elaborated how the hypertension screening should be performed. Please see lines 323-327 in page 11.

3. A BMI > 25 kg/m2 was found to be a risk factor for the development of hypertension and the authors further suggest weight reduction, exercise, and diet as part of obesity reduction; however, exercise was not found to be significant factor. What are the authors thoughts on this? 

Author's response: Though exercise was not found to be significant, we included as a way to reduce weight as it was identified as a significant factor (Zhu et al., 2022). Please see lines 288-291 in page 10.

4. With the males being underrepresented, could there be sampling error associated with the gender analysis? 

Author's response: There was no sampling error associated with the gender analysis, but there was scarcity of voluntarily consenting male participants. We also mentioned this in the limitation section.

5. Could kidney disease result from hypertension rather than kidney disease being a risk factor for hypertension?

Author's response: We found low renal function to be a significant risk factor of hypertension which is in agreement with Adeniyi et al (2017).

6. There are many reference errors throughout the manuscript that is listed as 'Error! Reference source not found'. Please ensure that all references are accurately noted. 

Author's response: We changed references from Endnote format to plain text.

Reviewer 2 Report

THE PAPER DESCRIBES THE RISK FACTORS FOR HYPERTENSION IN SOUTH AFRICAN HIV PATIENTS

THE PAPER IS WELL STRUCTURED BUT, CLARIFICATIONS ARE NEEDED BEFORE PUBLICATION

STARTING FROM PAGE 5 THE CITATION DO NOT APPEAR AND THEREFORE IT IS NOT POSSIBLE TO ASSESS THEIR RELEVANCE.

THE ANTIRETROVIRAL THERAPY ARE NOT CLEAR, THEY WOULD BE BETTER ILLUSTRATED.

AN ANALYSIS OF THE TYPES OF ANTIRETROVIRAL REGIMES AND THE RISK OF HYPERTENSION WOULD BE USEFUL.

IN THE END PART OF THE DISCUSSION THERAPEUTIC MANAGEMENT OF HYPERTENSION MUST BE MENTIONED (SEE  AIDS Rev 2017 Oct-Dec;19(4):198-211)

   (iT WOULD BE  USEFUL IF IN THE WORK THE ANTI-HYPERTENSIVE THERAPY IN ACT OR THE PERCENTAGE OF PATIENT IN ANTIHYPERTENSIVE THERAPY WERE UNDERTAKEN WAS SHOWED)

Author Response

  1. Starting from page 5 the citation do not appear and therefore it is not possible to assess their relevance.

Author's response: Thank you for all comments. We have noted that the referencing issue is software related and changed references from Endnote format to plain text.

  1. The antiretroviral therapy are not clear, they would be better illustrated.

Author's response: We used the names of antiretroviral therapy that was available in the patient files. We provided the full names of the regimen as footnotes under the tables and within the manuscript.

  1. An analysis of the types of antiretroviral regimes and the risk of hypertension would be useful.

Author's response: We included the analysis of the types of antiretroviral regimen and the risk of hypertension as part of figure 4(e), the risk table.

  1. In the end part of the discussion therapeutic management of hypertension must be mentioned (SEE AIDS Rev 2017 Oct-Dec;19 (4):198-211).

Author's response: Thank you so much, we revised as directed and added the reference. Please see lines 322-325 in page 11.

  1. It would be useful if in the work the anti-hypertensive therapy in act or the percentage of patient in antihypertensive therapy were undertaken was showed.

Author's response: The current study was looking at the time to the development of hypertension from the time of ART initiation among PLHIV, we excluded participants who had a history of developing hypertension.

Round 2

Reviewer 2 Report

the authors responded adequately to the criticisms that emerged in the previous version. I recommend checking the layout and some typos